# Evaluating Water and Carbon Retention in a Low-Order, Designed River Corridor

Jaclyn M. H. Cockburn [1,*] , Alex Scott [1] and Paul V. Villard [2]

1 Department of Geography, Environment & Geomatics, University of Guelph, Guelph, ON N1G 2W1, Canada
2 GEO Morphix Ltd., Campbellville, ON L0P 1B0, Canada
* Correspondence: jaclyn.cockburn@uoguelph.ca; Tel.: +1-519-824-4120 (ext. 53498)

**Abstract:** As urban residential areas expand into the areas around cities, especially in North America, these areas were previously forested or were converted to agricultural uses (e.g., cropping, grazing). Many of the pre-existing channels were modified prior to residential area expansion and required modification and/or restoration in order for development permits to be granted. These pre-existing channels are often low-order, semi-ephemeral streams with hydrological and geomorphological functions and provide aquatic-terrestrial habitat and ecological linkages. Once restored, these corridors provide important services to the entire river network related to flood-risk mitigation, sediment trapping, and are potential carbon (via particulate organic matter) sinks. This research evaluated water flow and carbon trapping within a low-order tributary of East Morrison Creek in Southern Ontario, Canada in the years immediately following construction. Water level records (5 September and 30 November 2019, and 1 April and 30 November 2020) show that even in its early development this new system was functioning efficiently. Sediment samples taken throughout the 2020 field season determined particulate organic matter was being stored, especially in features where flow was attenuated. Channel roughness imposed by large wood structures promote organic matter deposition within bed sediments and were expected to increase over time. These findings highlight the importance of spatial heterogeneity imposed by the design features used in this reach-scale restoration and serve as a valuable 'proof of concept' for future work along the urban-rural interface of expanding cities.

**Keywords:** channel restoration; post-construction evaluation; water retention; sediment storage; particulate organic carbon storage; aquatic-terrestrial ecosystem services; spatial heterogeneity

## 1. Introduction

As cities and urban areas expand, the previously forested or agricultural landscape (i.e., rural areas) are being (re)developed into residential, industrial, or commercial areas. This change in landuse impacts the existing watercourses [1,2], and usually coincides with substantial stormwater management needs, to mitigate these changes. Stormwater management systems can include a stormwater pond and channel networks connecting to existing or modified channels, with the overall goal of attenuating stormwater flow and improving water quality as the water moves downstream or connects to larger water bodies (e.g., streams, lakes). These stormwater management systems are vital to mitigating hydrological change due to increases in impermeable surfaces that enhances flooding, erosion risk and other associated hazards [3,4]. However, stormwater management systems can limit the aquatic ecosystem functionality (e.g., decreased carbon storage, biodiversity loss) due to the need to attenuate flow by altering hydrological connectivity and spatial heterogeneity [5–8].

Wohl et al. (2018) [9] discuss the importance of variability within a watershed and the potential for restored channels to provide more than a means of conveying water and mitigating flood risks. A watercourse that includes features that vary flow rates, velocities,

water depth can promote lateral and vertical connectivity, while maintaining effective longitudinal connectivity [10–15]. For example, ecosystem, geomorphic and soil attributes, and sediment and nutrient loading were evaluated in five restored urban systems in North Carolina, and found that in sites with increased floodplain connectivity, soil organic matter was higher and overall water quality improved [13]. Additionally, research on urban streams in California highlight the importance of limited incision rates; as riparian soils along these channels had greater organic matter content [16]. Prioritizing and purposefully designing, and amplifying connectivity in restoration works are vital in building resilient and effective water corridor designs.

Effectively replicating spatial heterogeneity and connectivity in a restored watercourse or as part of a stormwater management system is challenged further by space constraints and materials [17–19]. To mimic natural processes and ensure restored or newly designed watercourses function as close to a natural system as possible, design elements both instream and along the adjacent floodplain (e.g., pit and mound, in-line wetlands, woody debris, riffle-pool sequences) are included to induce variable velocities throughout the channel and encourage both lateral and vertical water exchanges (e.g., [11,20]). However, post-construction monitoring and evaluations of these projects typically focus on the function of the whole system and little is known about the effectiveness of individual design elements (e.g., online wetland) and lateral and vertical connectivity [21–25].

This study evaluated rainfall runoff attenuation and sediment-carbon retention within a low-order, restored reach of East Morrison Creek through various seasonal conditions following channel construction in 2019. Within East Morrison, innovative positioning of wetlands and wood structures (design features that promote water attenuation and sediment-carbon retention) aim to influence downstream water and material transport through active channel and floodplain retention, infiltration, evaporation, and evapotranspiration. Contributions from this research improve our understanding of current approaches to full corridor restoration projects along rural-urban transitions.

## 2. Study Site

The authors wish to highlight for readers that while there were many design options available for the study site featured in this work, the design that was implemented accounts for the unique permitting considerations and other site requirements present. For example, flood risk attenuation and water quality are critical permitting requirements is most jurisdictions, and designs need to be selected that match the local hydrological regimes. Additionally, species-at-risk, and species-of-concern, have specific habitat requirements that need to be addressed in water corridor design projects. Thus, what might work at one site may not be possible to implement in another due to ecological, regulatory, planning, and physical site constraints, that being said, the findings presented here are important 'proof of concepts' that could suit others with similar characteristics. Through 2018 and 2019, a stormwater system was designed and installed in Oakville, Ontario, Canada to accommodate residential development in previously agricultural and forested landscape (Figure 1; 43°29′26″ N, 280°16′2″ E). In association with the stormwater system project, stream corridor rehabilitation in the form of a channel design (conducted by GEO Morphix, Ltd., Milton, ON, Canada) took place along an upper tributary to East Morrison Creek. East Morrison Creek runs southeast, with forested and unused agricultural land to the northeast and northwest. The focused study area (Figure 1) had a reach-averaged bed gradient of 0.0041, channel width and depth ranged from 1.9 m to 2.3 m and 0.30 m to 0.45 m within pool sections and 1.40 m to 1.70 m and 0.15 m to 0.25 m within riffle sections (Figure 2). Local fill material (clay-sized to silt-sized material) overlies Queenston Formation (upper Ordovician shale, limestone, dolostone, and siltstone) [26]. During the stream corridor rehabilitation, an area three times the bankfull channel width was excavated from the floodplain and replaced with local fine-grained sands and gravels to promote water dispersal via infiltration and accommodate minor (~several metres) lateral channel adjustment.

Floodplain width ranged from 28 m to 31 m with a gentle downward slope between the outer floodplain extent and the channel banks (Figure 1).

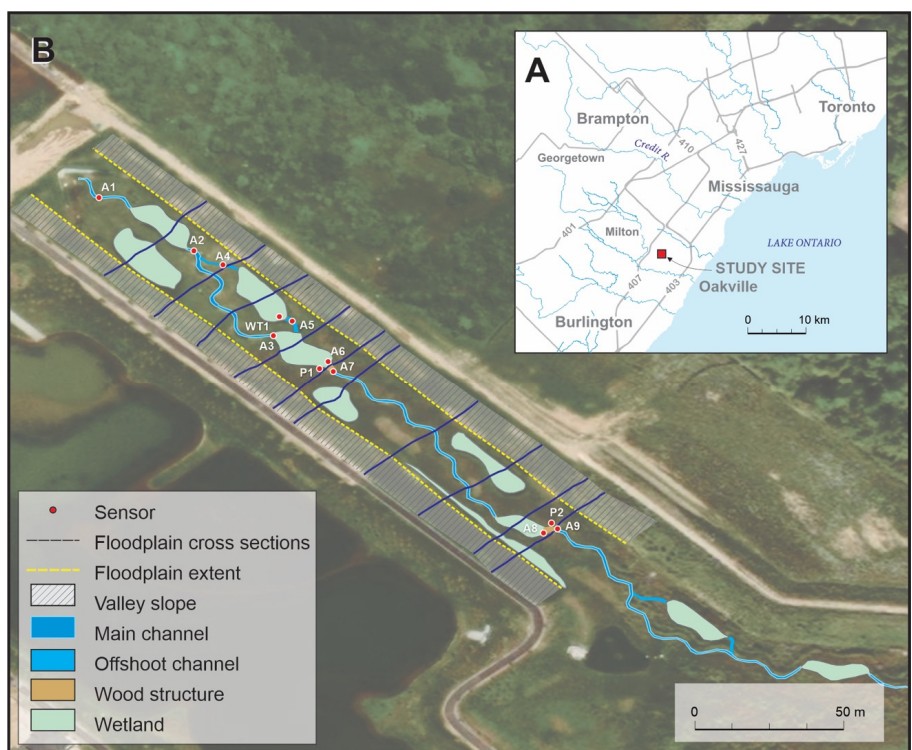

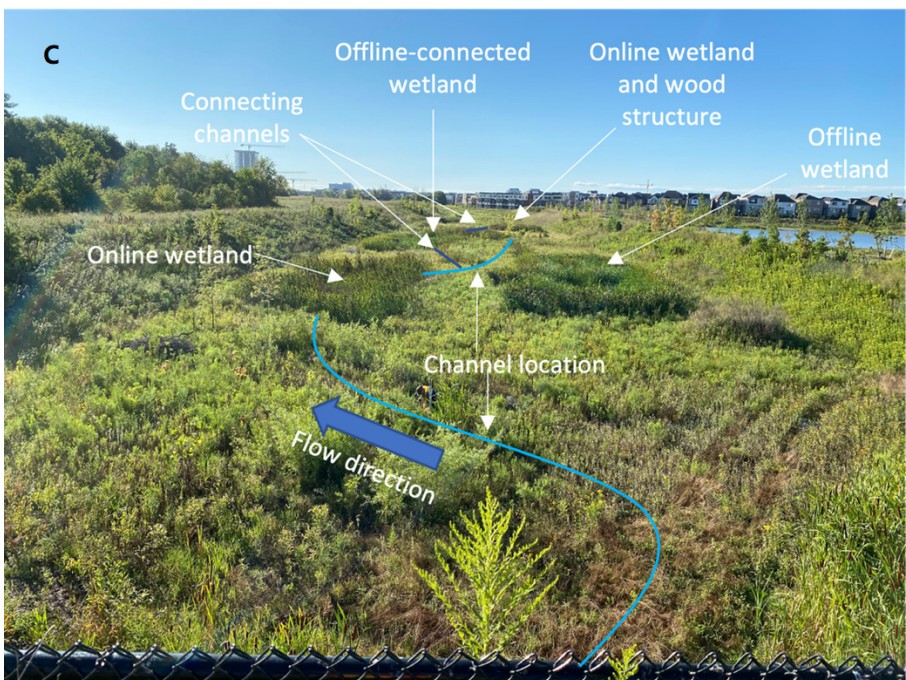

**Figure 1.** (**A**) Study area within the East Morrison Creek basin in southern Ontario, Canada. (**B**) Site planform view underlain by an UAV photo. Within the river corridor there is a main channel that contains online, offline, and offline-connected wetlands and wood structures. Sensor locations, floodplain lateral extent, and valley slope area are also shown. Flow is from northwest to southeast. Inset maps highlight locations with dense sensor locations. Map projection is NAD 1983 UTM Zone 17N. (**C**) Oblique photo from the upstream section looking downstream over the site. Wetland and channel locations are identified on the photo. Photo taken 10 June 2020.

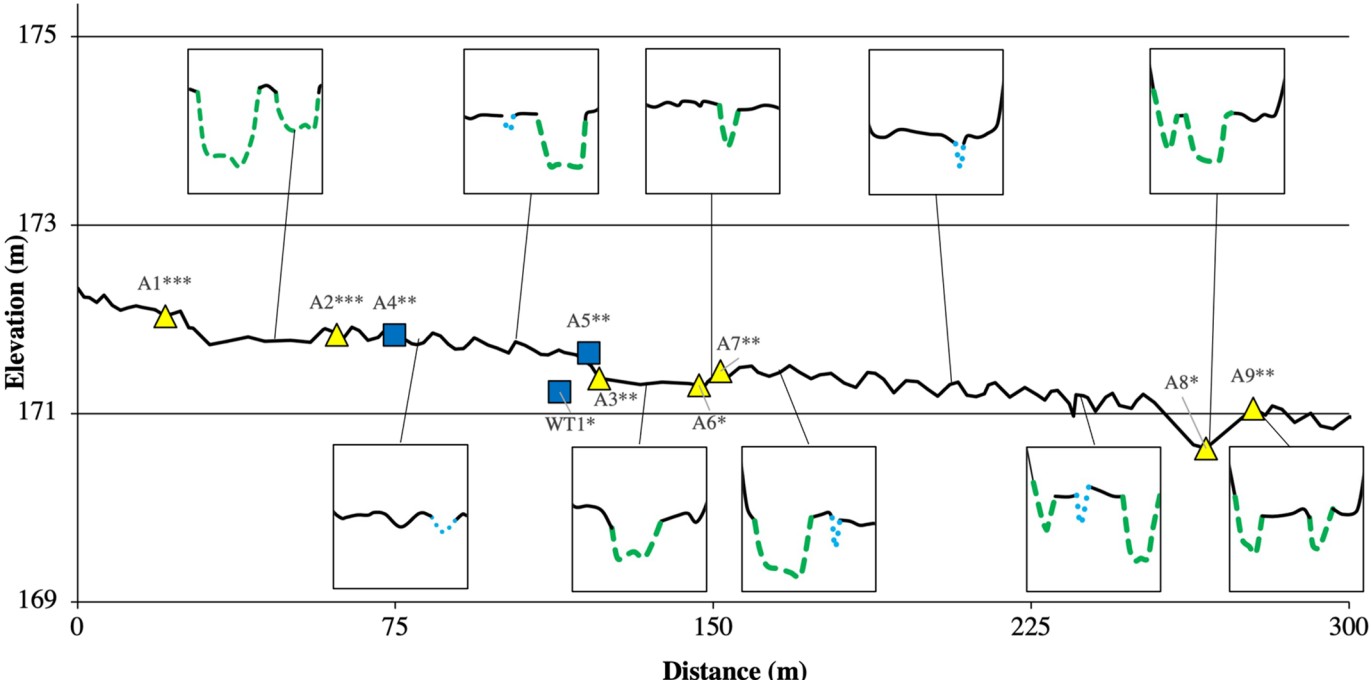

**Figure 2.** Longitudinal profile of the channel bed along the thalweg from upstream (left) to downstream (right). The thick black line denotes the channel bed along the thalweg. The thin black lines denote the intersection of each cross section with the main channel at the thalweg. Sensors located on the main channel are denoted by yellow triangles and sensors within the floodplain are denoted by dark-blue squares. Asterisk (*) suffixes denote sensors at features where flow was slower (e.g., pools), double asterisk (**) suffixes denote sensors at stations adjacent to slower flow (e.g., downstream of a pool), and triple asterisk (***) suffixes denote senor locations where flow was more efficiently conveyed (e.g., confined channels, minimal in-stream vegetation). The inset graphs show corridor cross sections, wetlands (within the floodplain and main channel) are denoted by a green dashed line and the main channel is denoted by a dotted light-blue line. Vertical exaggeration is 23.

Within the study reach, various design features promoted water retention. Wetlands were constructed throughout the reach and exist broadly in three types (Figure 1). Online wetlands are situated along the main channel. Offline wetlands are situated on the floodplain with no surficial connection to the main channel. Offline-connected wetlands are situated within the floodplain and connected to the main channel via offshoot channels activated during high-flow events. These wetlands were designed to promote water retention, elongate downstream hydro-periods, water dispersal, and sediment retention. Wetland surface area ranged from 150 m$^2$ to 341 m$^2$. Two wood structures are located within the channel just downstream of an online wetland (Figure 1). These wood structures were designed to provide increased in-channel roughness to aid in water retention, water dispersal and sediment trapping. The upstream and downstream wood structure surface area is approximately 4 m$^2$ and 9 m$^2$.

East Morrison receives water from the ~2 km$^2$ forested and disused agricultural areas lying to the northwest, according to the Ontario Flow Assessment Tool (OFAT) [27]. An upstream grass swale conveys water through a culvert into the reach. The upstream reach segment has two channel segments separated and immediately followed by online wetlands, an offline wetland, an offline-connected wetland (connected via offshoot channels), and a wood structure (Figure 1). A third online wetland and second wood structure are positioned at the downstream extent of the reach (Figure 1).

## 3. Methods

Post-construction monitoring began in early September 2019 (6 September) with seven monitoring stations (AT, A1, WT1, A6, A7, A8, A9), maintained by GEO Morphix Ltd. Station AT is a weather monitoring station measuring air temperature, rainfall, dew point, relative humidity, and wind speed. A1, WT1, A6, A7, A8 and A9 are surface water monitoring stations measuring surface water temperature and water level (Figure 1). Sensors were removed on 30 November 2019 and re-installed on 1 April 2020 and removed at the end of November 2020. On 10 June 2020, four more surface water monitoring stations were added (A2, A3, A4, A5). Each station measured surface water temperature and water level. Station A1 is the most upstream station and the channel cross-section at A1 has a low bankfull depth and width relative to others Figure 2). Stations A6 and A8 are each located within an online wetland and station WT1 is located within an offline-connected wetland. Stations A7 and A9 are each located just downstream of an in-channel wood structure, and typically ponded less frequently than stations A6, A8, and WT1 and more frequently than station A1. All sensors were removed on 30 November 2020.

HOBO Onset U20/U20L water level loggers (range: 30.6 m/−20–50 °C, accuracy: 1.0 cm/0.44 °C) and HOBO Onset U20-001 water level loggers (range: 9.0 m/−20–50 °C, accuracy: 0.5 cm/0.44 °C) were used to measure water level and temperature at 15 min intervals at all surface water stations. Sensors were placed 0.01 m above the bed, secured to a metal rod driven into the bed and encased in a stilling well made from PVC pipe (0.05–0.08 m diameter). The threshold water level value under which the sensor was assumed to be dry was determined on a station-by-station basis and verified during site checks. All water level values starting at site visits in which each station was dry through to the next rainfall event were removed from the dataset, and the maximum value was assigned as the threshold for the corresponding station. An additional HOBO Onset U20-001 water level logger measured air temperature at station AT. Rainfall was measured at 15 min intervals using a HOBO data logging rain gauge RG3 (range: 12.7 cmh$^{-1}$, accuracy: 2 cmh$^{-1}$) at station AT.

Two topographic surveys were conducted on 3 September and 4 November 2020, using Real Time-Kinematic (RTK) survey equipment (accuracy: 0.01 m). The surveys cumulatively produced 1850 three-dimensional topographic data points that defined valley, channel, wetland, and wood structure geometry, monitoring station locations, and reach gradient. Site visits were conducted bi-weekly between 8:00 and 13:00 from June to November 2020. Sensors were downloaded and relaunched at each station during each site visit. Water level, temperature, conductivity, and turbidity were discretely measured at stations with water present. Water depth was measured using a metre stick, water temperature and conductivity were measured using a YSI Pro30 handheld sensor (range: −5–55 °C/0–200 µScm$^{-1}$, accuracy: 0.2 °C/1 µScm$^{-1}$), and turbidity was measured using a YSI 9500 Ecosense Photometer (range: 5–400 FTU). Bankfull water level was measured at stations A1, A2, A3, A7, and A9 by lying a metre stick level with the top of each bank and measuring the vertical distance between the sensor bottom and the ruler. Bankfull water level at these stations was defined as the smaller of these two measurements. Bankfull water level at stations A6, A8, and WT1 was determined using topographic survey data by subtracting sensor elevation from the minimum top-of-bank elevation.

Sediment samples were taken five times through 2020 (29 June, 30 July, 15 September, 29 September, and 26 October). Three samples were taken at each location (where possible). In some instances, vegetation cover or deep water made it impossible to collect a sediment sample. Sediment samples were collected to determine percent mass organic matter (POM) using the loss-on-ignition method [28]. Samples were first oven-dried for 24 h at 60 °C. Each sample was placed into a pre-weighed crucible, weighed, placed in a muffle-furnace for 2.5 h at 550 °C, cooled, and then weighed again. Crucible mass was subtracted from pre- and post-furnace mass measurements. After subtracting crucible mass, post-furnace mass was subtracted from pre-furnace mass to determine loss-on-ignition. POM was determined by dividing sediment mass difference by pre-furnace sediment mass and multiplying by 100.

## 4. Results

### 4.1. Rainfall Variations 2019–2020

Over the short 2019 field season (6 September 2019, to 30 November 2019), total rainfall measured at the site was 230 mm. Over the 2020 field season (1 April 2020, to 30 November 2020), total rainfall measured at the site was 513 mm (Figure 3). Throughout both monitoring campaigns, there were no other major precipitation inputs other than rainfall (e.g., no measured snowfall, hail, etc.). Daily rainfall maximum was 41 mm on 27 October 2019, and 40 mm on 2 August 2020. Rainfall occurred more frequently in fall during the 2019 and 2020 field seasons (September, October) compared to the summer months. Historical, rainfall data from an Environment Canada weather station (ID 6155750) ~4.2 km NE of this study site, show how the 2019 and 2020 field seasons compare to longer term averages [29] (2009–2021) (Figure 3). April, June, and July 2020 rainfall totals were below the 2009–2021 Environment Canada station average and July 2020 rainfall was much lower than the average (Figure 3). August 2020 rainfall was much greater than average, related the 2020 August monthly rainfall total was the largest in 2009–2021 Environment Canada record.

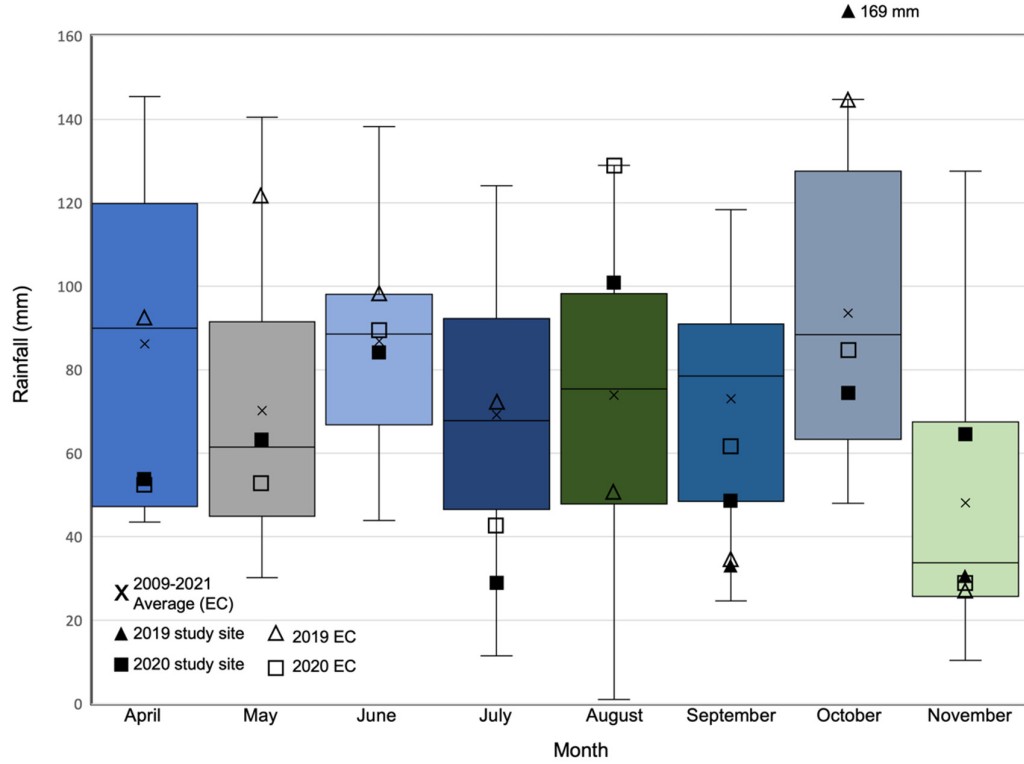

**Figure 3.** 2009–2021 monthly rainfall totals (x = average) for April–November from a nearby Environment Canada (EC) weather station (ID 6155750), and rainfall totals from the EC record (open symbols) and field site (closed symbols), for 2019 (triangle) and 2020 (square).

### 4.2. Water Temperature and Water Level 2019–2020

Water temperature and water level varied correspondingly to rainfall inputs throughout the season (Supplementary Material). Water temperature and water level data were reviewed to ensure they corresponded to times when water was present in the channel and limited to 1000 h and 1600 h to correspond with field visit times, and the Ontario Stream Assessment Protocol [30]. Generally, water level was highest immediately following rainfall and gradually decreased afterwards (Supplementary Material). Water level at stations A6, A8, and WT1 (situated within wetlands) was typically higher than water level at stations A7 and A9 (stations near wood structures) and station A3 (situated along a wetland margin) (Figure 1). Water level at stations A3, A7, and A9 was typically higher than water level at stations A1 and A2 (Figure 1).



The transition between wet (greater than 0.05 m water depth), shallow (wet, but less than 0.05 m water depth), and dry conditions at each station in response to rainfall events and dry periods illustrated how various elements within the study site respond to rainfall inputs (Figure 4). Wet/dry characterization was determined by extracting all time intervals over which each station was known to be dry, and the corresponding recorded water level during these times, then setting the threshold as the maximum of these values. Threshold values were no greater than 0.02 m. In early fall 2019 most stations were dry, with only WT1 and A8 having consistently wet conditions. As rainfall increased through the 2019 fall, all stations became wet and remained so other than station A1 (Figure 4). During the 2020 spring all stations remained wet for the entire season other than station A1, which dried at the end of April and varied between shallow and dry for the rest of the season (Figure 4). Spring moisture persisted through early summer 2020, however little rainfall over the season resulted in all stations drying by the end of July. A notable rainfall event in early August wetted all stations, after which hot conditions and no rainfall throughout the month led to all stations drying, with stations A1 and A2 drying first, then A7 and A9, and finally A6, A8, and WT1. In fall 2020, dry conditions continued through September. An early October rainfall event only wetted upstream stations, while stations A8 and A9 further downstream remained dry. Rainfall on October 15 wetted stations A8 and A9, and all stations remained wet through late October and November other than station A1, which varied between wet, shallow, and dry (Figure 4). Data availability was limited at multiple stations throughout the 2019 and 2020 field seasons due to sensor malfunction and human error when launching sensors. Conductivity, velocity, and turbidity observations were made throughout the season, with no notable trends observed [31].

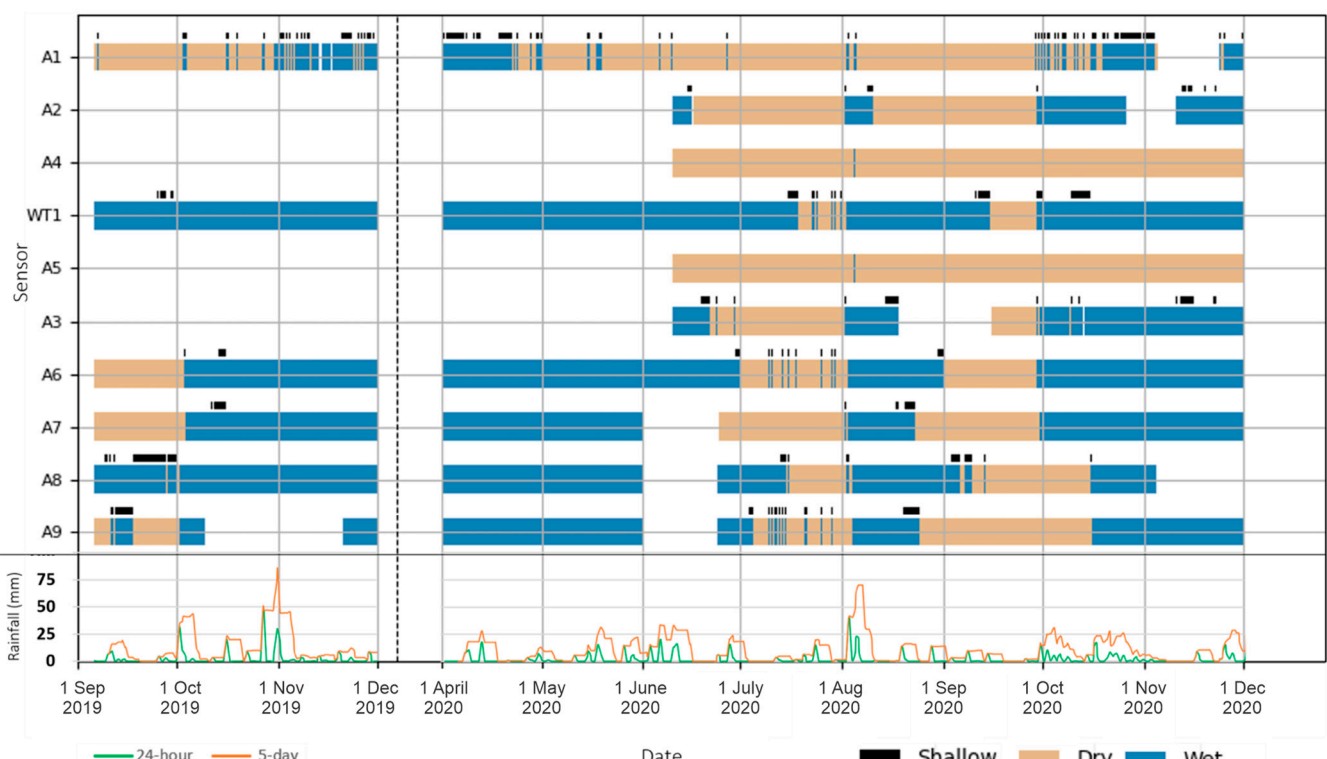

**Figure 4.** Top panel: Timeline bars of the 2019 and 2020 study periods indicating when a monitoring station is wet (blue) or dry (brown). Time steps are at 1000 h and 1600 h. Thin black bars above each timeline represent when the corresponding station water level was between 0.02 m and 0.05 m, signifying time periods when the station had shallow water. The dashed vertical lines indicate the break in the data between 30 November 2019 and 1 April 2020. Missing data are shown as gaps. Bottom panel: Cumulative rainfall over the previous 24 h (green line) and 5 days (orange line) for each 1000 h and 1600 h time step.

### 4.3. Organic Carbon Estimates in 2020

Percent organic matter (POM) was used as a proxy for particulate organic carbon stored within sediments [28]. Bed sediment samples collected to estimate percent organic matter (POM) throughout 2020 varied from ~5% to over 25% (Figure 5). Average sediment sample POM across all stations was much higher on July 30 compared to other sampling dates. Average POM was generally higher at stations A7 and A9 (situated near large wood structures) than station A6 (situated within a wetland), station A3 (situated at a wetland margin), and stations A1 and A2.

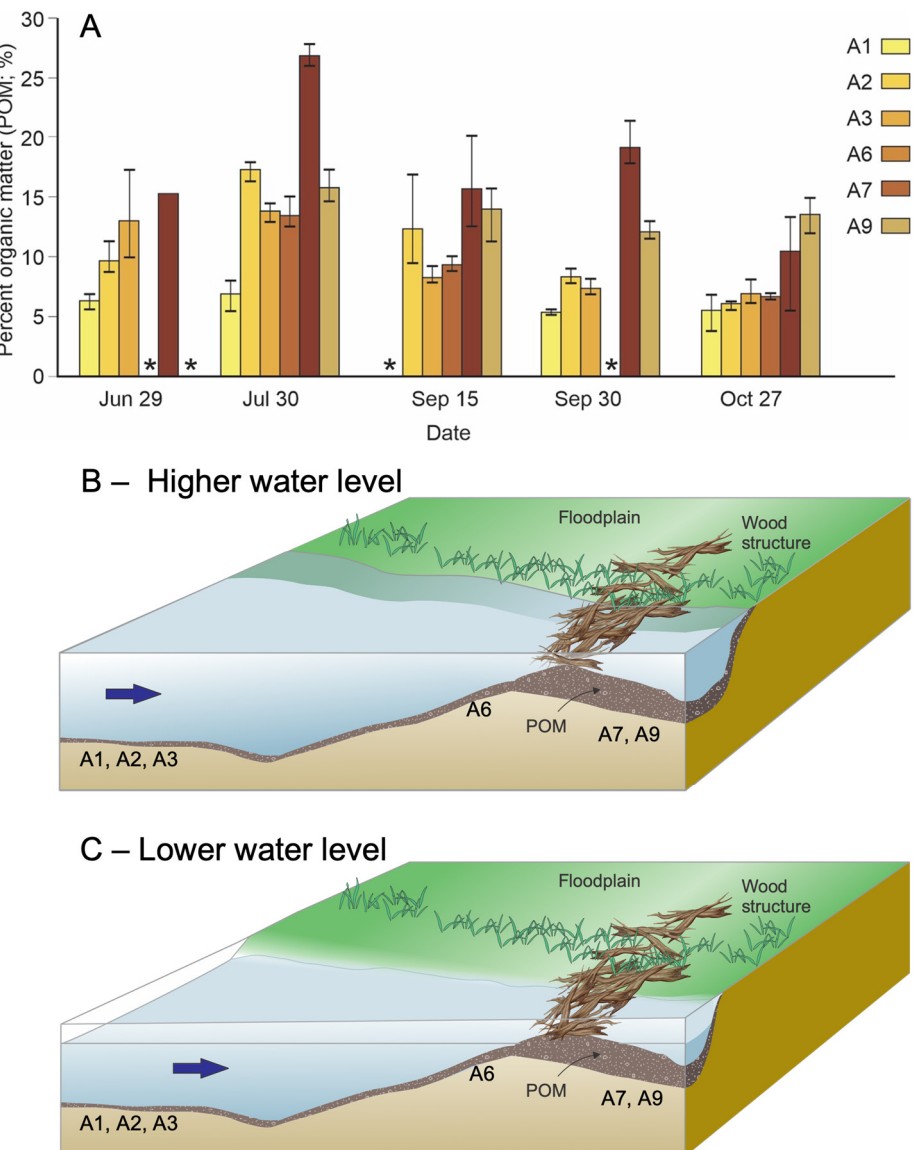

**Figure 5.** Percent organic matter (POM) in sediment samples taken from the bed at monitoring stations A1, A2, A3, A6, A7, and A9 throughout the 2020 field season (**A**). Error bars show POM range for each day sampled (three samples). Dates with an asterisk (*) indicate stations where no samples were retrieved. Schematic illustrations of the wood structures within the channel and corresponding (relative, not to scale) locations of the various stations for higher water levels (**B**) and lower water levels (**C**). Station A7 and A9 are immediately downstream of a wood structure and generally saw the highest amounts of stored organic material in the sediments sampled. A1 consistently had the lowest amounts, usually less than half of the amount observed at A7 or A9.

## 5. Discussion

### 5.1. Role of Spatial Heterogeneity in Attenuating Runoff

Spatial heterogeneity within the restored water corridors is important to promoting the various ecosystem services facilitated by water ways. Moreover, as landuse changes, water corridor restoration projects often need to meet permitting and stormwater management parameters. In the East Morrison Creek project, these demands were met by a unique design that promoted water attenuation in several features (e.g., wetlands (Figure 1)). Immediately following construction, the design was working as expected. During wet periods (e.g., fall 2019 and spring 2020), the water was moving through the system, all stations in the study were showing as having water present (Figure 4). October 2019 was the wettest in the past decade (Figure 3), with most of the rain coming over the last week of the month (Figure 4, bottom panel). In spring 2020, conditions in most small, headwater channels are wet, as cooler temperatures and potentially saturated ground following snowmelt ensure higher water tables (Figure 4; i.e., [15]). By mid-June 2020 only wetlands (online and offline), and limited channel stations recorded water presence (Figure 4). Through July 2020, further drying of the system was observed (Figure 4), such that during the largest rainfall of the 2020 field season, the system demonstrated its capacity. As flow within the system increased through rainfall events overall connectivity increased (Figure 4). Increased lateral connectivity was briefly observed as the connector channels (A4 and A5) were activated (Figure 1; Figure 4). Increased vertical connectivity was inferred from the persistent presence of water in the offline wetland WT1 (Figure 4). At the reach scale, these wetlands retain more water and thus minimized 'flashy' runoff responses to inputs, in addition to reducing the overall volume of runoff, these wetlands can promote water quality [23].

Understanding how these wetted areas expand and contract is essential to understanding how the entire system modifies connectivity and attenuate rainfall inputs water. In low-order southern Ontario streams like East Morrison, wetted area expansion was defined by a coalescence pattern [15]. In a coalescence expansion pattern, localized pools formed in topographically low points and coalesce to form a continuously wet channel. At a similar scale as Peirce and Lindsay (2015) [15] (i.e., from A1 to A7, approximately 50 m), coalescence also occurs at East Morrison, with A6, A8, and WT1 (being the topographic low points Figure 2). At slightly larger scales within East Morrison (i.e., from A1 to A9, approximately 150 m), downstream wetted area expansion occurs. In a downstream expansion pattern, upstream reach segments receive water, and channel segments progressively become wet in a downstream pattern. Peirce and Lindsay (2015) [15] conclude that downstream expansion did not occur within their study reach due to a low water table and unsaturated soils surrounding the channel. During the early August and early October 2020 rainfall events, downstream wetted area expansion was incomplete, as stations A1 to A7 became wet while A8 and A9 remained dry. An incomplete downstream expansion pattern suggests longitudinal connectivity may have been reduced and/or there was sufficient capacity in the upper part of the station to retain most of the inputs (Figure 4). Prolonged surface water presence at stations A6, A8, and WT1 following wetted area expansion suggests that water is being retained mainly at the surface within channel segments, with low gradients (e.g., ponding). Wetted area contraction within East Morrison is characterized by a disintegration pattern in which topographically high spots (e.g., A1, A2, A7 and A9) dry preferentially while low spots remain wet (e.g., A6, A8) (Figure 2, Figure 4). In this case, the mechanisms for water dispersal were infiltration into the subsurface and evaporation/evapotranspiration into the atmosphere from these topographically low spots, like that observed by Peirce and Lindsay (2015) [15]. Water retention within topographic low points (A6, A8, and WT1) during coalescence and non-downstream water dispersal during disintegration suggest that these topographically low segments within East Morrison provide longer-term water retention capabilities.

Wetland features are not the only way to promote connectivity, and at East Morrison, wood complexes (e.g., root wads, and wood structures), mimicking wood debris in natural channels reduces water velocity and flow by increasing in-channel roughness. Previous work has shown the importance of wood availability in biogeochemical processes as well as stream morphology (e.g. [4,9,16,19,22,32]). In East Morrison, the wood structures and online wetlands along the reach reduce water velocity and facilitate water retention within the channel, subsurface, and floodplain via overbank flow. The successful implementation of these design features in East Morrison and elsewhere (e.g., [13,23,33]) illustrate the ways in which this novel approach is transferable to other sites.

*5.2. Role of Spatial Heterogeneity in Promoting Carbon Storage*

In addition to water retention, variable flow rates (e.g., online wetlands and large wood structures) promote sediment retention, increasing organic matter (and thus carbon) storage within bed sediments [13,16]. POM within bed sediments at East Morrison varied throughout 2020, with stations positioned downstream of wood structures (A7 and A9) having the highest POM (Figure 5). Following reduced flow conditions during June and July 2020 (Figure 4), POM in bed sediments increased at most stations, with the greatest increase at A7 (Figure 5). Increased carbon retention downstream of wood structures was observed in other studies when compared to bar sediments in deciduous woodland rivers (e.g., [34]) and non-wooded meander bend sediment in mixed residential, grassland and woodland rivers (e.g., [35]). Wood structures promote carbon storage by inducing lee-side flow conditions suitable for POM deposition within bed sediments [35]. Moreover, POM sourced from the wood structures is transported into this depositional area and likely contributes to increased POM content [35]. The increased overbank flow frequency and subsequent sediment transport to longitudinally discontinuous floodplains induced by wood structures provides an additional, longer-term carbon retention environment [36]. Wood structures, whether they are purposefully constructed or occur through typical hillslope-channel processes, are important to facilitating spatial heterogeneity, aiding in water retention, and promoting carbon storage. Moreover, in addition to providing fish habitat, the purposeful capture and storage of organic carbon within low-order channels like the restored East Morrison Creek provide an important carbon sink on the landscape, that would otherwise not exist. Wood presence is incredibly important to channel design and/or riparian landscape conservation [19,33–35].

Urban systems are often overlooked in terms of their potential to serve as important carbon storage points within the fluvial landscape [9]. East Morrison Creek provides an important proof of concept in looking at the potential for carbon storage points in a restored channel. This underlines the importance of including wood as part of the design for restoration projects in urban systems, especially in systems where wood recruitment (e.g., supply) maybe be limited [33].

Lastly, these findings are specific to the period over which data were collected relative to channel construction (i.e., one to two years post-construction). It is expected that water and carbon retention within East Morrison Creek will change as the time post-construction increases. For example, POM was highest at stations A7 and A9, situated downstream of large wood structures, when compared to stations situated within the online wetland (A6). As vegetation at East Morrison matures, POM within wetland bed sediments sourced from in-channel vegetation will increase [7]. Increased roughness imposed by more mature and denser in-channel vegetation will also decrease flow rates and lead to higher sediment deposition, increasing the total mass of stored POM [12]. With time, it is expected that spatial heterogeneity will increase, thus continuing to promote flow attenuation and particulate organic carbon storage. Future research looking at 2-, 5-, 10 years post construction will help determine the trends and impacts of system maturation at the site but require time and additional funding to carry-out.

## 6. Conclusions

East Morrison Creek was designed to attenuate flow using features (i.e., wetlands and wood structures) to promote spatial heterogeneity. Runoff in response to rainfall events is retained within the main channel, as evident through the incomplete downstream expansion pattern occurring multiple times throughout the field observations. Flow attenuation occurs predominantly within wetlands, but also within topographically low channel segments. A disintegration wetted area contraction pattern suggests that water retained at the surface is infiltrated into the subsurface and/or is evaporated/evapotranspirated into the atmosphere following rainfall events. POM storage (within bed sediments) is highest within stations positioned downstream of wood structures. These findings are specific to the study period, which took place shortly after construction concluded, as such, the processes that facilitate water and carbon retention and the locations that disproportionately contribute to retention will likely change as time post-construction increases but demonstrates that this design is effective immediately. This is incredibly important as climate change and landuse change increase flood and other hazards along the urban-rural interfaces (e.g., [3,4]) and require creative and efficient solutions to mitigate. At East Morrison Creek, flood hazard mitigation is combined with improved water quality, and an opportunity to generate new carbon stores on the landscape.

**Supplementary Materials:** The following supporting information can be downloaded at: https://www.mdpi.com/article/10.3390/land11122256/s1, Figure S1: Total daily rainfall, water temperature and level over the 2019 and 2020 field seasons.

**Author Contributions:** Author contributions are as follows. Conceptualization, J.M.H.C., P.V.V., A.S.; methodology, J.M.H.C., A.S.; formal analysis, A.S., J.M.H.C., P.V.V.; investigation, A.S., J.M.H.C.; resources, P.V.V., J.M.H.C.; data curation, J.M.H.C., A.S.; writing—original draft preparation, A.S., J.M.H.C.; writing—review and editing, J.M.H.C., A.S., P.V.V.; visualization, A.S., J.M.H.C.; supervision, J.M.H.C., P.V.V.; project administration, J.M.H.C.; funding acquisition, J.M.H.C., A.S. All authors have read and agreed to the published version of the manuscript.

**Funding:** This research was funded by Canadian Foundation for Innovation, grant number 31341. A.S., received a Mitacs Research Training award in 2020. The APC was waived.

**Data Availability Statement:** This article is based on the MSc. Thesis research conducted by A.S. and available here. Some data are proprietary, please contact the corresponding author for details (jaclyn.cockburn@uoguelph.ca).

**Acknowledgments:** AS thanks University of Guelph Department of Geography, Environment and Geomatics and Mitacs Research Training Award (2020) for financial support. JC acknowledges infrastructure funding support from the Canadian Foundation for Innovation (CFI). Field and technical support provided by GEO Morphix staff is greatly appreciated. Marie Puddister in the Department of Geography, Environment and Geomatics at University of Guelph provided production support. Reviewer comments were extremely helpful in clarifying and enriching this study—Thank you for the effective and constructive suggestions.

**Conflicts of Interest:** The authors declare no conflict of interest. The funders had no role in the design of the study; in the collection, analyses, or interpretation of data; in the writing of the manuscript; or in the decision to publish the results.

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
