# Peer review of "Evaluating Water and Carbon Retention in a Low-Order, Designed River Corridor"

_land, doi:10.3390/land11122256_

Round 1

Reviewer 1 Report

This is a fine and relevant manuscript! I have a major question and some minor issues for improvement.

Major question: You argue a lot with the advantage of carbon storage. I wonder if this is really true (or if this is so remarkable). If the POM would not be stored in the East Morrison Creek, it would flow into Lake Ontario and would be stored there. Thus my question: Does the design for East Morrison Creek make any difference for the total carbon storage?

If you find a positive answer to this question, you should include the arguments in 5.2. , where you currently just take it for granted (in lines 332/333)

If you don´t find a positive answer, you should scale down the carbon storage argument in the article.

And here my minor issues for improvement:

- Probably "connectively" in line 59 should be replaced by "connectivity"

- explain the "the unique permitting considerations and other site requirements present" (line 80) in more detail, because this will be helpful for readers who would like to apply your results to their cases

- add two photos of the wood structures, one at low water and one at high water

- lines 289-292 are difficult to understand, and there is a mistake in line 289

- the discussion should include reflections on the transferability to other sites

Author Response

The authors are very appreciative of the review and comments – THANK YOU!

In particular the major question: “You argue a lot with the advantage of carbon storage. I wonder if this is really true (or if this is so remarkable). If the POM would not be stored in the East Morrison Creek, it would flow into Lake Ontario and would be stored there. Thus my question: Does the design for East Morrison Creek make any difference for the total carbon storage?” is a really important point, and we revised to strengthen this point in the resubmitted paper (added citations in the last paragraph of section 5.1, added text and citations at the end of the first paragraph of section 5.2, and in the conclusion).  This additional text was used to highlight that if these designed structures weren’t included, carbon storage wouldn’t happen at all – so it isn’t like it would end up in Lake Ontario (Line 374-378 in the revised submission).

Response to minor issues:

  • Line 59 in original submission “connectively” changed to connectivity
  • Line 80 in original submission – additional details provided in lines 81-85.
  • Photos of wood structures --- unfortunately, we do not have effective photos of the wood structures (debris jams for this site). In the revised manuscript, we have modified Fig 5 to include higher and lower water level schematics.
  • Line 289 in the original submission (now Line 297) – mistake corrected (deleted the unnecessary word “occurred”)
  • Lines 289-292 in the original submission, now Lines 297-302 were revised to clarify the types of connectivity observed and the evidence that supports this (went from one long sentence to several sentences)
  • Reflection on transferability (inclusion/incorporation at other sites) was included at the end of section 5.1 (Lines 340-342)

Reviewer 2 Report

In this article, the authors have made a very meaningful and interesting study that evaluated water and carbon retention in a Low-Order, designed River Corridor. These findings highlight the importance of spatial heterogeneity imposed by the design features used in this reach-scale restoration and serve as a valuable concept for future work along the urban-rural interface of expanding cities. However, there are still some problems for minor revision before publishing in Land.

1. In order to enrich the introduction and let readers understand the current research status, the authors should also add some other literature achievements to illustrate the importance of the research.

2. Please divide the 4 Results into different sections (4.1......, 4.2......) according to the logical characteristics of the paper.

3. In the discussion or conclusion of this article, the authors can appropriately increase the shortcomings of this study and the possible future research directions. If the authors put forward some meaningful suggestions from the perspective of ecological environment or land use according to the research results, which will enrich the research content.

4. From references in this article, there are many references repeated citations by the authors. In fact, the references related to the subject in this study are very rich, and the authors can add other references according to the difference between the introduction and the discussion.

Author Response

The authors appreciate the care and attention provided by this review, it was extremely helpful in improving the revised submission.

Below is a point-by-point response to the issues highlighted by this review, starting with the numbered point from the reviewer, followed by our response in italicized font.

  1. In order to enrich the introduction and let readers understand the current research status, the authors should also add some other literature achievements to illustrate the importance of the research.

Several references were added to the introduction to better situate the importance of the present study within the literature with respect to challenges associated with climate change, and issues uniquely associated with urban systems. Additionally, references from the Discussion were included into the introduction where appropriate.

– Kumar et al 2022 & Hou et al 2020 – discuss the potential hazards in urban areas due to climate change when stormwater systems are not adequately constructed

– Newcomer et al 2012 – highlights the effectiveness of various treatment options in mitigating nutrient outputs from restored system.

  1. Please divide the 4 Results into different sections (4.1......, 4.2......) according to the logical characteristics of the paper.

The Results section was subdivided into different sections as suggested

  1. In the discussion or conclusion of this article, the authors can appropriately increase the shortcomings of this study and the possible future research directions. If the authors put forward some meaningful suggestions from the perspective of ecological environment or land use according to the research results, which will enrich the research content.

This comment aligns with a similar comment from the other reviewer. Additional text was included at the end of section 5.1, end of the first paragraph in 5.2, and with the additional text at the end of the Conclusion paragraph. Future research directions were added at the end of section 5.2 (Lines 395-397) Shortcomings were previously addressed in the last paragraph of Section 5.2.

  1. From references in this article, there are many references repeated citations by the authors. In fact, the references related to the subject in this study are very rich, and the authors can add other references according to the difference between the introduction and the discussion.

This was a helpful comment – thank you. We have revised the references cited between these two sections to include references used throughout the discussion were introduced in the discussion where appropriate.